# Transfer of Stress Resilient QTLs and Panicle Traits into the Rice Variety, Reeta through Classical and Marker-Assisted Breeding Approaches

**DOI:** 10.3390/ijms241310708

**Published:** 2023-06-27

**Authors:** Saumya Ranjan Barik, Arpita Moharana, Elssa Pandit, Abhisarika Behera, Ankita Mishra, Shakti Prakash Mohanty, Shibani Mohapatra, Priyadarsini Sanghamitra, Jitendriya Meher, Dipti Ranjan Pani, Vijai Pal Bhadana, Shiv Datt, Chita Ranjan Sahoo, Reshmi Raj K. R., Sharat Kumar Pradhan

**Affiliations:** 1ICAR-National Rice Research Institute, Cuttack 753006, India; saumya_bt06@yahoo.co.in (S.R.B.); arpiarpita22k@gmail.com (A.M.); abhisarika_in@yahoo.com (A.B.); ankitamishra547@gmail.com (A.M.); evenpunk22@yahoo.co.in (S.P.M.); baanibt@gmail.com (S.M.); p.sanghamitra1@gmail.com (P.S.); jmehercrri@gmail.com (J.M.); reshmirajkr@gmail.com (R.R.K.R.); 2Department of Biosciences and Biotechnology, Fakir Mohan University, Balasore 756020, India; 3College of Agriculture, Odisha University of Agriculture & Technology, Bhubaneswar 751003, India; chitta.ouat@gmail.com; 4Environmental Science Laboratory, School of Applied Sciences, KIIT Deemed to be University, Bhubaneswar 751024, India; 5ICAR-National Bureau of Plant Genetic Resources, Base Center, Cuttack 753006, India; paninbpgr@gmail.com; 6ICAR-Indian Institute of Agricultural Biotechnology, Ranchi 834003, India; bhadanavijai@gmail.com; 7Indian Council of Agricultural Research, Krishi Bhavan, New Delhi 110001, India; shivdatts1@gmail.com

**Keywords:** *Sub1*, *Pup1*, *GW5*, stress resilient, yield improvement, marker-assisted breeding

## Abstract

Reeta is a popular late-maturing high-yielding rice variety recommended for cultivation in the eastern Indian states. The cultivar is highly sensitive to submergence stress. Phosphorus deficiency is an additional constraint for realizing high yield. The quantitative trait loci (QTLs), *Sub1*, for submergence and *Pup1* for low phosphorus stress tolerance along with narrow-grained trait, *GW5* were introgressed into the variety from the donor parent, Swarna-Sub1 through marker-assisted breeding. In addition, phenotypic selections for higher panicle weight, grain number, and spikelet fertility were performed in each segregating generation. Foreground selection detected the 3 target QTLs in 9, 8 and 7 progenies in the BC_1_F_1_, BC_2_F_1_, and BC_3_F_1_ generation, respectively. Recurrent parent’s genome recovery was analyzed using 168 SSR polymorphic markers. The foreground analysis in 452 BC_3_F_2_ progenies showed five pyramided lines in homozygous condition for the target QTLs. No donor fragment drag was noticed in the *Sub1* and *GW5* QTLs carrier while a segmentwas observed in the *Pup1* carrier chromosome. The developed lines were higher yielding, had submergence, and had low phosphorus stress-tolerance alongwith similar to the recipient parent in the studied morpho-quality traits. A promising pyramided line is released in the name of Reeta-Panidhan (CR Dhan 413) for the flood-prone areas of Odisha state.

## 1. Introduction

Rice is life for millions of farmers and rice based traders in India and many Asiatic countries. Rice cultivation for consumption provides livelihood for millions of farmers in India. Importance of rice is known to people even from its use in rituals for birth, marriage and death and other ceremonies. Rice though mainly supplies carbohydrate but compounds such as quality proteins, many minerals, vitamins, specific oils, and dietary fiber along with few useful antioxidants are available in the grain [1,2,3]. The crop is cultivated from very high elevations to a below sea level and is highly diverse in nature. Rice is cultivated in about 160 Mha in the world. The annual value from the rice is about 206 billion USD, which gives about 17% of the total crop value globally [4]. Rainfed rice is the practice in about 45% of the total rice cultivated area [5]. The production and productivity of the crop is now highly challenged by many climate change-related adverse effects [6]. The distribution pattern of monsoon rain in India during the last few decades confirms about the variability of rainfall distribution which causes instable rice production in the rainfed ecology. The variability in monsoon rainfall may invite stresses like drought, submergence, salinity, and mineral deficiencies. Few stresses are interrelated and the effects on rice yield may be very high. 

The area under rainfed rice is 22 Mha, of which 90% is located in eastern region of India [7]. Flash flood is a common adverse effect faced by the farmers in this region for and suffers a huge crop loss by growing high yielding and submergence sensitive varieties [8,9]. The occurrence of flash floods is common in rainy season in India, particularly in the eastern states of the country. Low rice production in the region is mainly due to the cultivation of sensitive varieties those are affected by submergence stress. Rice variety, Reeta (CR Dhan 401) is a popular variety for the late maturity group but susceptible to submergence stress under flash flood situation. A total crop failure occurs if the crop is exposed to two weeks submergence stress due to flash flood. But, the panicle of Reeta variety gives more grains/panicle and bears heavy panicle than the variety, Swarna-Sub1. However, the grains of Reeta are bold type with more gain width than Swarna-Sub1. Varieties built-in with submergence tolerance QTL, *Sub1*, confers tolerance for approximately two weeks submergence stress [10]. Gene-based and closely linked markers are available for marker-assisted selection for transfer of the *Sub1* QTL. This tolerance trait has been incorporated into many high yielding varieties including Swarna using QTL introgression [7,11,12,13,14,15,16]. The transfer of the target gene from an improved source to recipient variety may contribute very less undesirable genetic effects from the donor variety [17].

Phosphorous (P) is an essential major nutrient for rice plant growth, development and higher production. P-deficiency is a global problem and tentatively 5.7 billion hectares of cultivated areas are with phosphorus deficient [18]. The soil phosphorus availability is low or medium in about 80% of the districts in India [19,20]. Direct seeded rice is a common practice for rainfed ecology that faces a deficiency in phosphorus very frequently [7]. Among the major nutrients, P is the second limiting nutrient for rice cultivation. About 30 to 65 percent of total soil phosphorus is in organic and unavailable forms to the plant. Indian farmers usually neglect application of required quality of this fertilizer due to it’s high cost in our country [21]. In addition, fertilizer runoff may leads to creation of pollution in water bodies. Therefore, rice cultivation with tolerance to low soil phosphorus level is very much needed for the farmers in India and abroad. The trait is controlled by a major QTL, *Pup1* which is mapped and located on chromosome 12 providing phenotypic variance of >70% tolerance to phosphorus deficiency in soil [15,22]. The full *Pup1* region comprised of 68 genes of which *PSTOL1* QTL is the most probable candidate gene that confers low-P stress tolerance in rice [23]. The other region within the *Pup1* QTL may provide a supportive role for better phosphorus uptake Rice varieties with low soil P stress tolerance will be P efficient and will produce more at a lower input cost. Adaptation of low p-stress tolerant varieties/cultivars requires neither additional cost nor major changes in the cropping system. The *Pup1* introgressed rice varieties developed by marker-assisted breeding programs showed very good results in the field trials [24,25]. Thus, popular varieties need to be pyramided with low P-tolerance QTL, *Pup1* that can grow and yield well even under low P-availability.

The narrow grained rice is more preferred and a market driven trait in recent years in India. Though there is impressive growth in rice production and productivity in recent years, the demand of this type of rice is growing upward in the country. Enhancement of yield is also regulated by the grain weight and grain dimension QTLs viz., *GS3*, *GW2*, *GS5*, and *GW5* [9]. Reduced grain width is associated with the QTL, *GW5* and the effects of the QTL were consistent under multiple environments. Deletion of a nucleotide segment in the *GW5* region resulted in wide-grained genotypes in most *japonica* and *indica* rice [26]. The donor parent, Swarna-Sub1 exhibits narrow grain type and carries the QTL, *GW5* [5]. The variety Reeta is a late maturing variety which produces a wide-grained kernel. This investigation aims at the development of pyramided lines carrying *Sub1, Pup1*, and *GW5* (narrow grain) QTL for submergence, low phosphorus tolerance and narrow-grained rice along with panicle traits improvement in the popular variety background, ‘Reeta’.

## 2. Result

### 2.1. Validation of the Donor and Recipient Parents for the Target Traits

The target QTLs namely *Sub1*, *Pup1*, and *GW5* controlling the traits were validated for their presence by comparing the donor versus recipient parents before beginning the hybridization and selection activities. The presence of target QTLs, *Sub1* for submergence tolerance, *Pup1* for low phosphorus tolerance, and *GW5* for yield component grain weight were confirmed in the donor parent (Appendix A). The recipient parent, Reeta was a high-yielding popular variety but deficient in these QTLs. The gene-based and tightly linked molecular markers for *Sub1*, a direct marker for *Pup1*, and the yield component QTL, *GW5*, were used to validate and track the target QTL in the parental and backcross-derived lines (Appendix A; Appendix A). A parental polymorphism survey was performed by using 728 simple sequence repeat markers covering all the chromosomes. A total of 168 polymorphic markers between the two parents were deployed for background screening (Appendix A).

### 2.2. Marker-Assisted Selection in BC_1_F_1_ Progenies

Reeta was hybridized with ‘Swarna-Sub1′, and 250 F_1_ seeds were obtained. The hybridity in F_1_ plants was confirmed by genotyping the hybrid plants using the *Sub1* specific marker. One true F_1_ plant was crossed with the recipient parent, Reeta, and a total of 265 BC_1_F_1_ seeds were generated. A total of 138 backcross generation 1 seeds were raised and the remaining 127 BC_1_F_1_ seeds were stored as backup seeds. Foreground screening was performed in all the 140 BC_1_F_1_ plants using the 7 selected molecular markers for the QTLs, *Sub1*, *Pup1,* and *GW5* (Figure 1).

The screening results from 138 BC_1_F_1_ progenies of the backcross revealed the presence of *Sub1* QTL in 65 derivatives detected by the markers Sub1 A203 and Sub1BC2 (200 bp and 240 bp). Tracking of the *Pup1* QTL by the markers Pup1K46, RM28073 and RM28102 showed its presence in 25 individuals out of a total of 65 derivatives with submergence tolerance. Screening for the presence of *GW5* Swarna allele (narrow-grain) controlling the grain width and weight was detected in 9 individuals to carry all the 3 target traits. Background screening was performed in the 9 BC_1_F_1_ foreground-positive progenies by using 168 SSR markers. Out of these 9 individuals, the progeny carrying the maximum recipient genome content was selected for the next backcross. The recurrent parent’s genome content in those 9 progenies varied from 64.88% to 80.95% with an average value of 76.91% (Table 1). The backcross derivatives RSS45 and RSS223 showed the recurrent genome content of 80.24% and 80.95%, respectively. However, higher panicle weight, grain number and spikelet fertility were better in RSS45 than RSS223. The BC_1_F_1_ line generated from RSS45 was backcrossed with the recipient parent, Reeta, to obtain BC_2_F_1_ seeds.

### 2.3. Marker-Assisted Selection in BC_2_F_1_ Generation

One hundred and thirty-five BC_2_F_1_ plants were grown in the field for selection. The 3 target QTLs were tracked by foreground selection using gene-specific and linked markers. The genotyping results of 135 BC_2_F_1_ individuals showed 64 positive progenies for *Sub1* QTL. These individuals were checked for the presence/absence of the *Pup1* QTL using markers Pup1K46, RM28073, and RM28102. A total of 27 positive plants carrying *Pup1* QTL were identified by tracking the QTL. Those 27 plants were checked for the presence of the *GW5*Swarna allele (narrow-grain) using a gene-specific marker. Eight plants with all the desired QTLs were identified for further background selection (Figure 2). Background screening for recovery of the recipient parent genome in those 8 identified plants containing the 3 targets QTLs varied from 81.54% to 92.26% with an average of 89.06% (Table 1). The plant RSS45-83 showing 92.26% of the recipient variety Reeta genome content, was used for the next BC_3_ backcrossing.

### 2.4. Marker-Assisted Selection in BC_3_F_1_ and BC_3_F_2_ Generations

The BC_3_F_1_ seeds were generated by crossing BC_2_F_1_ plant no. RSS45-83 and the recurrent parent ‘Reeta’. A total of 156 BC_3_F_1_ seeds were generated and raised for molecular screening by foreground and background selections. The genotyping results for the target QTL, *Sub1*, were positive in 65 progenies. Those 65 *Sub1* carrier plants were genotyped for checking the presence/absence of the *Pup1* QTL. Within this population, 27 plants were positive for 3 *Pup1* markers. A total of 30 positive plants were screened for *GW5* QTL. This analysis identified 7 plants positive for *GW5* and further genotyped for background screening (Figure 3). The background analysis using 168 SSR markers in these 7 plants detected 92.86 to 95.24% recurrent parent’s genome recovery, with an average of 94.30% (Table 1). The highest recurrent genome containing plant RSS45-83-27 was self-bred, and 22.5 g seeds were produced for further evaluation in BC_3_F_2_ generation. A total of 452 self-bred seeds were raised, and the rest seeds were kept as backup seeds. All the plants were subjected to foreground screening, of which 96 plants were identified to be homozygous for the *Sub1* allele. All the *Sub1*-homozygous plants were screened homozygosity for *Pup1* QTL carrying plant. A total of 19 plants showed the presence of *Pup1* in the homozygous condition. Those plants were screened for the presence of *GW5* in homozygous conditions. Finally, 5 plants were identified to carry all the 3 targets QTL in homozygous condition (Figure 4). Seeds of these five pyramided lines were used for further evaluation in the subsequent generations for various morphological and quality traits. Cluster analysis with agro-morphologic and quality traits showed distinct clusters of Reeta with 2 pyramided lines, RSS45-83-27-54 and RSS45-83-27-415 while the donor parent was placed in a separate cluster (Figure 5A). Additionally, a dendrogram was generated by using the alleles detected with the SSR markers, which grouped the developed pyramided and parental lines into two groups. Group 1 is further divided into two subgroups with the 3 pyramided lines (RSS45-83-27-54, RSS45-83-27-526, and RSS45-83-27-648) and the recipient parent in one cluster and the other 2 pyramided lines in a separate group (Figure 5B). Five pyramided lines were accommodated in cluster I along with the recipient parent ‘Reeta’, while the donor parent remained in cluster II. The backcross-derived lines in cluster I were found to form different subclusters based on the 14 agro-morphologic traits studied but were similar to the recipient parent ‘Reeta’ for the majority of the studied morphological and quality traits. The pyramided lines RSS45-83-27-54, RSS45-83-27-235, RSS45-83-27-415, RSS45-83-27-526, and RSS45-83-27-648 were almost similar in terms of genome recovery among themselves and with recipient parent ‘Reeta’ except grain size.

### 2.5. Analysis of Genome Introgression on the Carrier Chromosomes of the Pyramided Lines

The background analysis for recipient genome recovery and genetic drag linked to the donor segments were assessed using 168 background and 7 foreground markers. The markers were carefully selected for all the chromosomes to obtain maximum coverage in background screening. The foreground analysis detected five BC_3_F_2_ pyramided lines carrying the 3 target QTLs in the homozygous condition in the progenies. The *Sub1* carrier on chromosome 9 showed no linkage drag of the donor fragment on both sides of the marker RM8300 and Sub1BC2 in all five NILs (Near Isogenic Lines) (Figure 6). Also, the *GW5* (narrow-grain) carrier chromosome present on the chromosome 5 showed no drag of the donor segment in all pyramided lines. However, a donor segment was detected in between the marker RM28073 and RM28102 in all the pyramided lines (Figure 6).

### 2.6. Evaluation of the Pyramided Lines for Submergence Tolerance

Seven genotypes including five pyramided lines carrying target QTLs were evaluated under the controlled submergence screening tank for confirmation of the submergence tolerance trait in the pyramided lines. Two weeks of complete submergence stress was given to the test genotypes by exposing the materials to 1.5 m water depth after 18 days of transplanting. After one week of de-submergence, all five pyramided lines showed regeneration ability from 85 to 95% while the donor parent ‘Swarna-Sub1’ showed regeneration of 95% (Figure 7). No regeneration was found in the sensitive parent ‘Reeta’. The pyramided lines RSS45-83-27-54, RSS 45-83-27-235, RSS 45-83-27-415, and RSS45-83-27-648 showed high regeneration ability under control testing (Figure 8). The pyramided lines showed high regeneration abilities of about 90% regeneration ability.

### 2.7. Evaluation of the Pyramided Lines under Low Phosphorus Stress

Molecular screening was performed to identify better lines for efficient phosphorous uptake. Both gene-specific as well as flanking markers were used to detect the presence of *Pup1* locus in the 5 pyramided lines during screening in the segregating generations (Appendix A). Swarna-Sub1 and Reeta were used as positive and negative checks. RM28073 and RM28102, the two closest markers to *Pup1* QTL [24]; one gene-based marker, Pup1K46; *Pup1* negative and positive checks were used in the pot study. The samples at dough stage of the plants were collected and dried in a hot air oven at 80 °C for recording of dry weight. The oven dried plant materials were chopped and grounded in a Willey mill and stored in wide-mouthed Stoppard bottles. After suitable sub-sampling, the samples were analyzed for total phosphorus by Vanadomolybdate yellow color method. The genotypes showing positive response for all the 3 markers employed were taken as positive for *Pup1* locus. Based on this criteria, 5 genotypes were observed to be positive for *Pup1* QTL (Figure 6). Pup1K46 being a dominant marker associated directly with the *PSTOL1* gene showed the expected amplicon of 523bp in both derived and pyramided lines and the tolerant parent used in this study (Appendix A). The results from the pot study also clearly showed that the pyramided lines carrying *Pup1* QTL showed better P-uptake under normal and deficient p-condition. The –ve check lacking the *Pup1* QTL showed relatively less uptake of phosphorus in the P-deficient condition than normal condition (Table 2).

### 2.8. Evaluation of the Pyramided Lines for Agro-Morphologic, Yield and Grain Quality Traits

The pyramided lines carrying submergence and low phosphorus tolerance along with yield component QTL in the background of the Reeta variety were evaluated for various traits during the wet seasons in 2020, 2021, and 2022. The pyramided lines were compared with both the parental varieties Reeta and Swarna-Sub1. The recipient parent ‘Reeta’ produced a pooled mean grain yield of 4.38 t/ha under typical shallow lowland conditions. The pyramided lines RSS45-83-27-54, RSS45-83-27-235, RSS45-83-27-415, RSS45-83-27-526, and RSS45-83-27-648 produced more yield than the recipient parent, Reeta (Table 3). However, the agro-morphologic traits of all of the pyramided lines were not similar to that of the parent, Reeta. The target morphologic traits controlled by the yield component QTL viz., no. of primary branches, spikelet fertility, and panicle weight finally influencing grain yield in the pyramided lines were observed to be almost similar within each subcluster (Table 3; Figure 8). Much of the grain quality and the cooking characteristics of the recipient parent such as milling (%), head rice recovery (%), gel consistency, amylose content (%), and alkali spreading value were retained in the pyramided lines similar to the recipient parent (Table 3). The placement pattern of the parents and the pyramided lines in the quadrants of the genotype-by-trait biplot diagram constructed based on 14 agro-morphologic, yield, and component traits over three years showed similarity among the pyramided lines (Appendix A). The pyramided lines were found in the first and second quadrants along with the recipient parent ‘Reeta’. The developed lines closer to each other were almost similar in grain yield, grain quality, and the other studied parameters (Appendix A). The best-line was evaluated in various parts of the country through All India Coordinated Rice Improvement Programme and showed superiority and stability in performance compared to the check varieties in Odisha state. The variation observed for the first principal component was 61.75%, while 15.05% was explained for the second component.

## 3. Discussion

Marker-assisted backcrossing (MAB) is a technology far superior over the conventional methods of selection and breeding in precision and efficiency for improvement of user friendly crop plants. Integration of a molecular marker for selection of segregating progenies enhances the accuracy in transfer of a desired trait into a recipient variety through backcross breeding program. This technology helps in identification of recombinants exhibiting the least amount of linkage drag and thus decreases selection time. In the realm of climate change induced disaster-prone rice agro-ecosystem, MAB assumes great significance by decreasing selection times of major stress-resilient gene(s) involved and pyramiding them successfully. In the present study, rice cultivar ‘Reeta’ is a popular variety, but long maturity duration increases its chances of being inundated by flash floods that results in oxidative stress and soil P-deficiency detriment of grain yield. MAB breeding approach adopted herein has achieved a great distinction in developing pyramided-lines, which exhibited submergence and low phosphorus tolerance, and a higher yield without compromising the main features of the farmer-friendly variety. The 3 target QTLs were simultaneously transferred into it. In addition, breeding the variety duration was reduced through MAB compared to the classical backcross breeding approach. The backcrossing was continued up to 3 backcrosses and then one selfing generation were performed to transfer the target QTLs into the variety. Thus, the essential lowland features lacking in variety were improved precisely in less time duration with negligible genetic drag. Variety development through marker-assisted breeding by precise transfer of genes and with a shorter duration has been reported in rice improvement by several workers previously [27,28,29,30,31]. Our work has added a new dimension to these illustrious examples because of the successful introgression of not only stress resiliency but also concomitantly improving grain yield. 

Earlier successful gene transfers and pyramiding results in rice were published in rice crop [14,28,29,30,31,32,33,34]. Further, this pyramiding study of QTLs conferring tolerance for submergence and low phosphorus stresses including improvement of yield component QTL was clearly different from earlier gene pyramiding publications. The earlier publications on gene pyramiding of bacterial blight resistance along with submergence tolerance into rice varieties, namely improved Tapaswini and improved Lalat have been reported, but the recipient varieties belonged to the mid-early maturing group and were not suitable for lowlands [35,36]. Hence, yield improvement along with submergence and low phosphorus tolerance through gene pyramiding is a novelty and typical example of gene stacking for a variety of rainfed lowland rice ecology. In contrast to the plethora of publications on the development of cultivars through pyramiding of resistance genes for insects and diseases in rice [29,30,31,32,33,34,35,36,37], the present research targets the development stress tolerance like submergence and P-deficiency tolerance. The pyramided lines carrying the *Pup1* locus showed not much difference in P-uptake in the plant tissue under normal and deficient phosphorus in the pots. The –ve check showed higher P-uptake under normal P-level but low uptake under deficient conditions (Table 2). Such a result confirms expectations of differential response between the low P-tolerant and P-susceptible varieties. There are instances of a similar correlation between the *Pup1* allele expression and P-uptake in studies reported earlier [20,22].

Majority of the crop breeding strategies for biotic and abiotic stress tolerance are based on a single resistant gene introgression into plants and they are short-lived. By contrast, pyramidization of multiple resistant genes into a single plant from different sources is durable and ensures stability. Besides, the success of trait correction is achieved in the shortest period of time for agricultural sustainability, as cited in previous publications [14,32,33,34,35,36,37]. However, in the present work, the developed pyramided lines carrying the target QTLs viz., *Sub1*+ *Pup1* + *GW5* along with recipient parents’ genome content of >95% were clearly different from what was reported previously. It is possible that an undesirable drag from the donor genome may come in progenies as additional unlinked loci in the backcross generations [31]. In our investigation, such effects were detected but from the elite donor source into pyramided lines while transferring the *Sub1*, *Pup1* and *GW5* QTLs into the Reeta variety background. The linkage drags are depicted in the graphical representation of genotyping data on the chromosome carrying the target QTLs (Figure 7). A very low linkage drag was noticed while transferring 3 target QTLs because the donor parent was an elite variety. In previous studies also low linkage drag was noticed while using elite donor sources and using more background markers [29,30,34]. The donor parent used in this study was Swarna-Sub1 which is a popular variety, and therefore, the expected drag may not show any undesirable effects in the pyramided lines (Figure 7). Similar findings were also reported by many workers which recommend the use of an improved variety as the donor that results in less or no undesirable drag compared to the wild and landraces sources [28,29,30,35,36,37,38,39,40].

In fact, there were a few pyramided lines showing similarity to the recipient parent in many traits. The dendrogram generated based on the traits indicated grouping of the pyramided and parental lines into mainly 2 clusters with similarity within the clusters (Figure 6A). The pyramided and recipient parents were found in quadrants I and II in the biplot diagram drawn based on 14 studied morpho-quality traits, showing minor variations among the pyramided lines (Appendix A). Evaluation of the pyramided lines for yield and grain quality traits showed higher yields in pyramided lines viz., RSS45-83-27-54, RSS 45-83-27-235, RSS 45-83-27-415, RSS 45-83-27-526, and RSS45-83-27-648 than the recipient parent, Reeta (Table 3). The transfer of traits and achieving similar or better yield in the pyramided lines were also reported earlier in some gene-pyramiding publications [1,28,29,30,35,36,37,38,39,40].

In our work, the pyramided lines were closer to each other with the recipient parent, Reeta in the trait-biplot diagram, while the donor parent was far away and placed in another quadrant. Therefore, the similarity of the pyramided and recipient lines was quite similar and no linkage drag was observed from the donor parent in the transfer of the target genes. In addition, the yield, quality, and other morphological traits in a few pyramided lines were better than the recipient parent (Table 3). It is clear from the genotyping results of background analysis that there was an accelerated recovery of the recipient parent’s genome in a few pyramided lines than the expected value in the backcross generations. Also, it revealed that the transfer of *Sub1*, *Pup1* and *GW5* QTLs into one genetic background may not show antagonistic effects for yield and related traits. 

## 4. Materials and Methods

### 4.1. Plant Materials and Breeding Program

In the breeding program, Swarna-Sub1variety bearing *Sub1*, *Pup1*, and *GW5* QTLs, with agronomic traits submergence tolerance, low P (phosphorus) deficiency-tolerance, and grain size respectively, was used as the donor male parent. The recipient parent, Reeta (CR Dhan 401) was a high-yielding variety, but vulnerable to submergence stress. The varieties were obtained from the gene bank of the ICAR-National Rice Research Institute, Cuttack, India, and grown in the screening tanks and crossed in the wet season of 2017 as per the scheme depicted for marker-assisted breeding (Figure 9). According to the protocol, one true F_1_ plant was hybridized with the recipient parent during the dry season of 2018 to generate BC_1_F_1_ generation seeds. True hybridity was checked using the direct *Sub1*-marker, Sub1A203, and a co-dominant marker, RM8300. The BC_1_F_1_ seeds were grown, and the progenies were screened for the target genes for submergence tolerance (*Sub1*), P-deficiency tolerance (*Pup1*), and grain size (*GW5*) by using the established molecular markers (Appendix A). A selection was performed to get the Swarn-Sub1 grain width (*GW5*). In the background selection, progenies of the BC_1_F_1_ generation carrying the 3 target QTLs were screened using the polymorphic markers. Among the foreground positive progenies, the lines containing highest genome of the recurrent parent was hybridized with the recipient parent, Reeta to get BC_2_F_1_ seeds. BC_2_F_1_ seeds were harvested during the dry season of 2019. The background analysis of the BC_3_F_1_ progenies was performed during the same season. The BC_3_F_1_ plant population containing the highest recipient genome content along with two major target QTLs *Sub1* and *Pup1* were self-bred during the wet season of 2019. The BC_3_F_2_ progenies were genotyped to search for the presence of homozygosity for the two major target QTLs and the recipient parent’s *GW5* QTL during the dry season of 2020. Foreground positive plants exhibiting higher panicle weight, grain number and panicle branching were also checked in each segregating generation. Phenotyping of these QTLs was performed and evaluated during the wet seasons in 2020 and repeated in the years 2021, and 2022.

### 4.2. Genomic DNA Isolation, Polymerase Chain Reaction, and Marker Analysis

Genomic DNA content was isolated following the standard extraction protocol [41]. PCR reaction was performed following the procedure used in our previous publications [3,42,43]. The information regarding chromosome number, position, and sequence of the primers used in the polymerase chain reaction are presented in Appendix A. Seven gene-specific and tightly linked markers for the two target QTLs and four recipient QTLs were used in foreground selection (Appendix A). The markers used in this study were taken from earlier publications [6,9,17,20,21,25,26]. A total of 728 SSR markers available in public domain were used for the study of polymorphism between the two parents. The polymorphic markers detected were used for background selection (Appendix A). Agarose gel electrophoresis was used to segregate the amplification products obtained from PCR reactions. The images were recorded in a gel documentation system (SynGene, Cambridge, UK). Data analysis and dendrogram construction were performed following the standard publications [44,45,46]. Graphical Geno Types (GGT) Version 2.0 software was used to construct the genome recovery graph of the recipient parent in the pyramided lines based on the SSR marker data [47].

### 4.3. Screening for Submergence Tolerance

The BC_3_F_4_ generation pyramided lines and parents were transplanted (3 weeks old seedlings) in the screening tank of ICAR-NRRI, Cuttack, during the wet seasons in 2020 and 2021. The screening trial was laid out in a randomized complete block design (RBD) with three replications/entries accommodating a population size of 66 plants/entry. The experiment materials were transplanted at a spacing of 15 × 20 cm^2^ by providing three rows/entry. Two weeks of complete submergence stress was given by exposing the materials to 1.5 m water depth after 18 days of transplanting. De-submergence was performed just after the completion of the 14-day stress period, and subsequently, regeneration was assessed one week after de-submergence. The data recording and scoring of the genotypes were performed following the standard procedures [17].

### 4.4. Phenotyping for Phosphorus Uptake

The pyramided and parental lines were evaluated under deficient and normal soil phosphorous in the pot experiment. The experiment was laid out in a randomized block design with 3 replications for both phosphorus levels. In the deficient condition, the P-content of the collected low-P upland soil was 6.5 mg/kg while the P-content in the normal soil was 13.3 mg/kg after the enrichment of single super phosphate fertilizer. Five genotypes observed to be homozygous for *Sub1*, *Pup1*, and *GW5* QTLs along with their two parents were grown in the pots. Nitrogen was applied in three equal splits viz., basal, active tillering, and panicle initiation. A full dose of phosphorus and potassium was applied as a basal application. Seeds were direct seeded in thepots and after germination thinned later to maintain two seedlings per pot. The samples at dough stage of the plants were collected and dried in a hot air oven at 80°C for recording of dry weight. The oven-dried plant materialswere chopped and ground in a Willey mill and stored in wide-mouthed Stoppard bottles. After suitable sub-sampling, the samples were analyzed for total phosphorus. One gram of powdered sample was taken and subjected to triacid acid (nitric acid (ISOCHEM): sulphuric acid (ISOCHEM): perchloric acid (SDFCL) of 3:2:1 ratio) digestion. The solution was filtered and the volume was made up to 100 mL using distilled water. Five ml of the triacid extract was pipetted out into a 25 mL volumetric flask. Five ml of Barton’s reagent was added and the volume was made up to mark with distilled water. The development of the yellow color was observed after 30 min and the intensity of color was measured in a photoelectric colorimeter using a blue filter (470 nm) after adjusting the transmittance of the meter to 100 with a blank. The color was stable for 24 h. The concentration of phosphorus in the solution was deduced from the standard curve from which the percentage of phosphorus content of the sample was calculated.

### 4.5. Evaluation of the Pyramided Lines

The seedlings of 25-day-old pyramided Reeta background lines carrying *Sub1*, *Pup1*, and yield QTLs were transplanted along with the parents during the wet seasons in 2020, 2021, and 2022. A plot size of 12 m^2^ was provided for each entry, with 40 plants per row, at a spacing of 15 × 20 cm^2^, and planted in RBD with three replications in the research farm of NRRI, Cuttack. The data for ten plants for morpho-quality traits viz., plant height, panicles/plant, panicle weight (g), number of filled grains, total spikelets, number of primary branches, secondary branches, and number of tertiary branches per panicle, grain length, grain breadth, 1000-grain weight, milling (%), head rice recovery (%), and amylose content (%) from each entry and replications were recorded. Plot yield and days to 50% flowering were recorded on a whole plot basis. The standard protocols for head rice recovery and gel consistency [48] were adopted. Amylose content in the grains of the pyramided and parental lines was estimated following the standard procedure [49].

### 4.6. Statistical Analysis

The recorded morpho-quality traits of the pyramided and parental lines were analyzed using SAS 2008, version 9.2 [50]. The Principal Component Analysis (PCA) for the pyramided and parental lines was performed by using multivariate analysis (Past Software version 4.03) data of the 15 morphological traits. A scatter plot was generated by using two major components: Principal Component 1 (PC1) and Principal Component 2 (PC2). The Eigen value and percentage of variance were generated by the interaction of a variance–covariance matrix. The interaction between all morphological traits was depicted through a biplot graph in the matrix. All the plots and results of PCA were generated as per the standard procedure followed in earlier publications [51,52,53,54].

## 5. Conclusions

The near-isogenic lines, RSS45-83-27-54, RSS 45-83-27-235, RSS 45-83-27-415, RSS 45-83-27-526, and RSS45-83-27-648 carrying *Sub1*, *Pup1* and *GW5* were submergence and low-P stress tolerant with higher yield than the recipient parent, Reeta. The higher yield recorded from the pyramided lines might be due to the accumulation of additional yield and stress tolerance QTLs. In addition, no yield penalty happened due to the interaction of these QTLs in the pyramided background. The grain quality and the cooking characteristics of the recipient parent such as milling %, head rice recovery %, gel consistency, amylose content (%), and alkali spreading value along with yielding ability were retained in a few pyramided lines. Hence, the elite pyramided lines in the background of the popular variety ‘Reeta’ may serve as potential donors of QTLs possessing *Sub1* + *Pup1* + *GW5* in future breeding programs. Based on our innovative research, a promising near-isogenic line has been released in the name of Reeta-Panidhan (CR Dhan 413) for cultivation in the flood-prone areas of Odisha state. This study established the application of marker-assisted selection for transferring abiotic stress tolerance and for enhancing yield in rice.

## Figures and Tables

**Figure 1 ijms-24-10708-f001:**
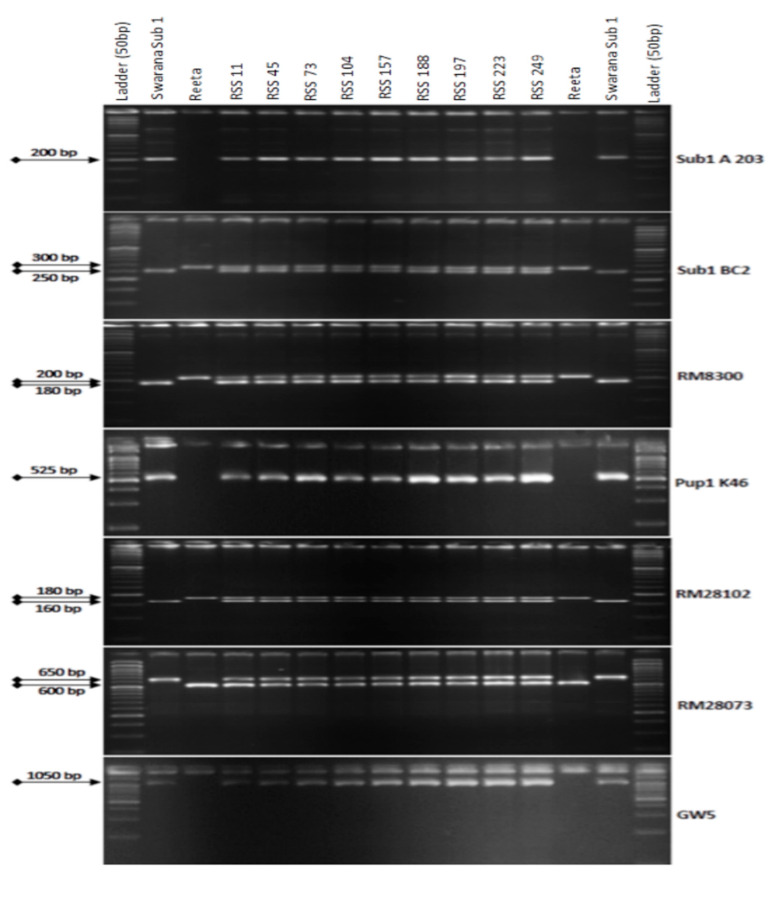
PCR amplification of markers Sub1A203 and Sub1BC2 linked to submergence tolerance; Pup1K46, RM28073 and RM28102 linked to low phosphorus tolerance genes along with grain yield marker *GW5* in BC_1_F_1_ progenies. L: Molecular weight marker (50 bp plus ladder) and lanes on the top of the gel indicate BC_1_F_1_ progenies (RSS: Reeta Swarna-Sub1).

**Figure 2 ijms-24-10708-f002:**
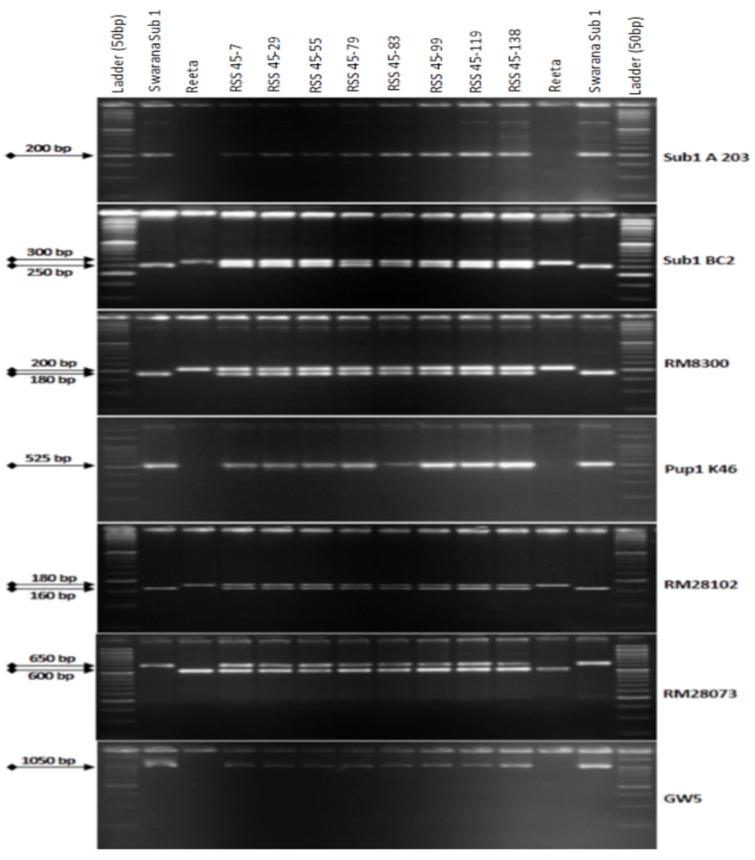
PCR amplification of markers Sub1A203 and Sub1BC2 linked to submergence tolerance; Pup1K46, RM28073 and RM28102 linked to low phosphorus tolerance genes along with grain yield marker *GW5* in BC_2_F_1_ progenies. L: Molecular weight marker (50 bp plus ladder) and lanes on the top of the gel indicate BC_1_F_1_ progenies (RSS: Reeta Swarna-Sub1).

**Figure 3 ijms-24-10708-f003:**
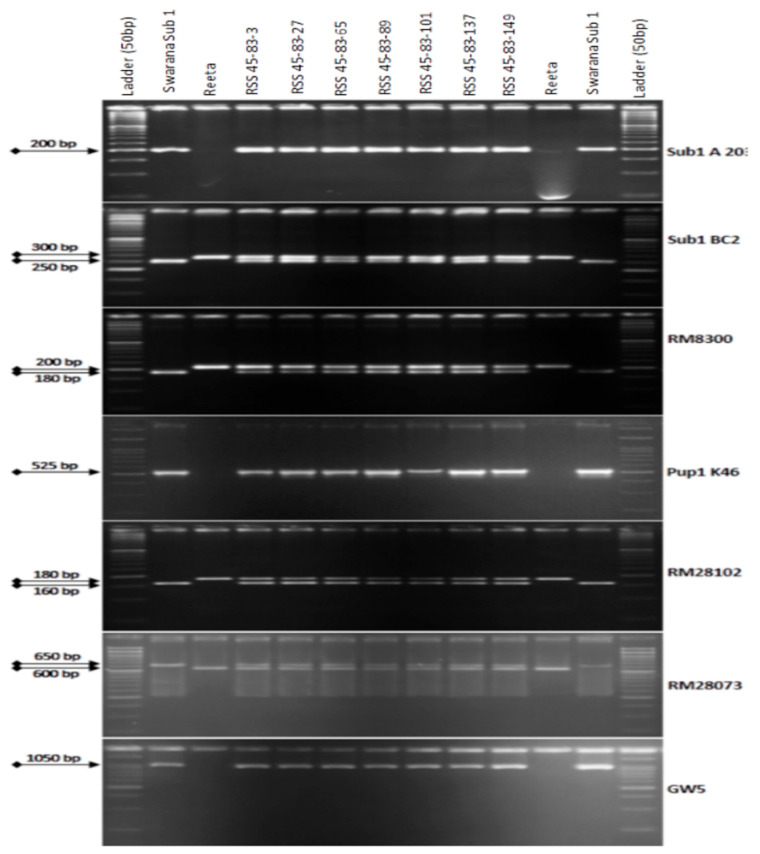
PCR amplification of markers Sub1A203 and Sub1BC2 linked to submergence tolerance; Pup1K46, RM28073 and RM28102 linked to low phosphorus tolerance genes along with grain yield marker *GW5* in BC_1_F_1_ progenies. L: Molecular weight marker (50 bp plus ladder) and lanes on the top of the gel indicate BC_3_F_1_ progenies (RSS: Reeta Swarna-Sub1).

**Figure 4 ijms-24-10708-f004:**
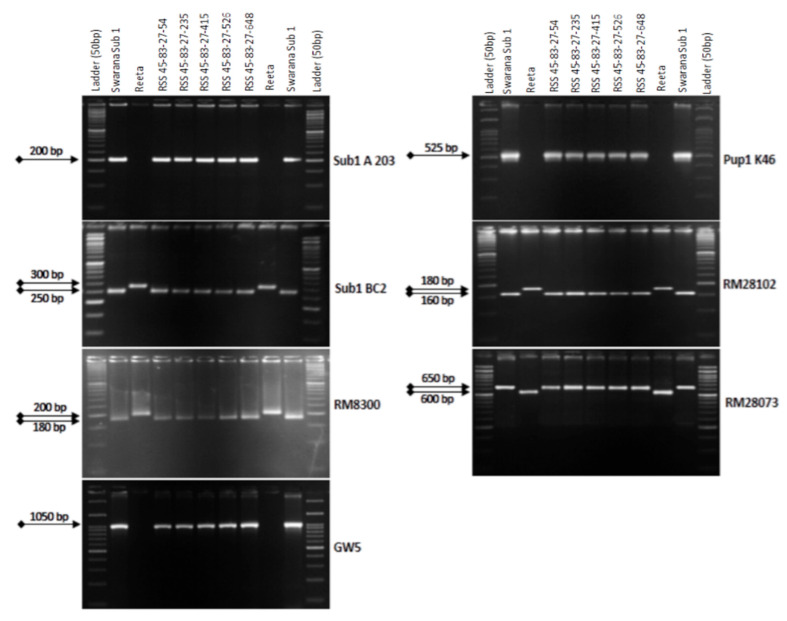
PCR amplification of markers Sub1A203 and Sub1BC2 linked to submergence tolerance; Pup1K46, RM28073 and RM28102 linked to low phosphorus tolerance genes along with grain yield marker *GW5* in BC_1_F_1_ progenies. L: Molecular weight marker (50 bp plus ladder) and lanes on the top of the gel indicate BC_3_F_2_ progenies (RSS: Reeta Swarna-Sub1).

**Figure 5 ijms-24-10708-f005:**
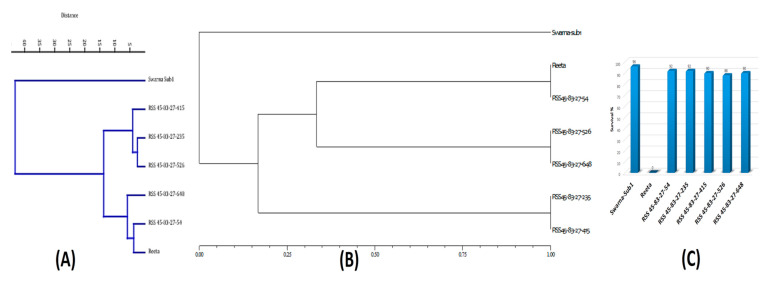
Seven pyramided lines of Swarna-Sub1 and Reeta along with parents in (**A**) dendrogram showing relatedness based on the 14 morphologic and quality traits; (**B**) Dendrogram showing the genetic relationship between lines based on 7 microsatellite markers and (**C**) % contribution of recurrent genome in the pyramided lines.

**Figure 6 ijms-24-10708-f006:**
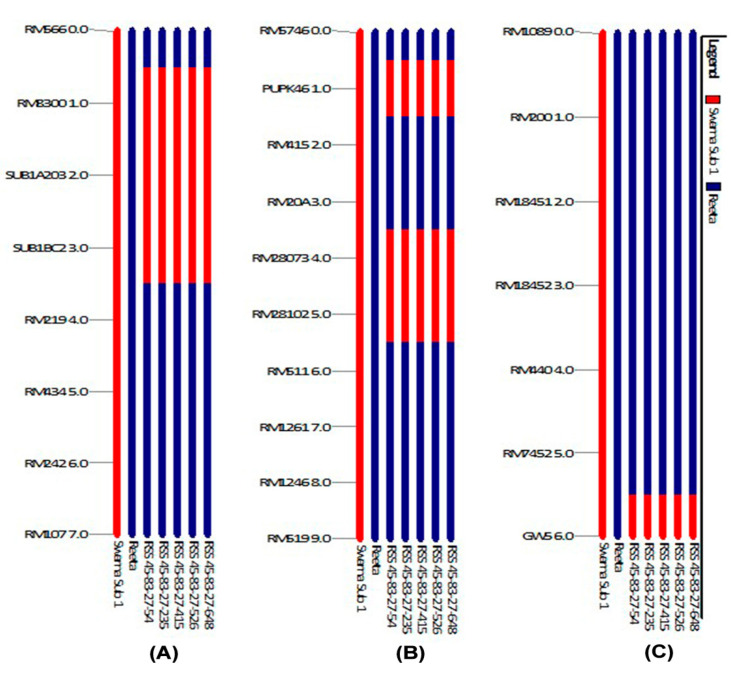
Analyses of QTLs stacking in carrier chromosomes associated with submergence tolerance, low phosphorus tolerance and yield component QTLs in the 5 pyramided lines (**A**) *Sub1* QTL on the carrier chromosome 9 (**B**) *Pup1* and (**C**) *GW5* (narrow-grain) yield component QTLs on the carrier chromosome 5 present in the BC_3_F_3_ progenies of Reeta/Swarna-Sub1. The numbers indicate the pyramided lines, 1. RSS45-83-27-54, 2. RSS45-83-27-235, 3. RSS45-83-27-415, 4. RSS45-83-27-526, and 5. RSS45-83-27-648.

**Figure 7 ijms-24-10708-f007:**
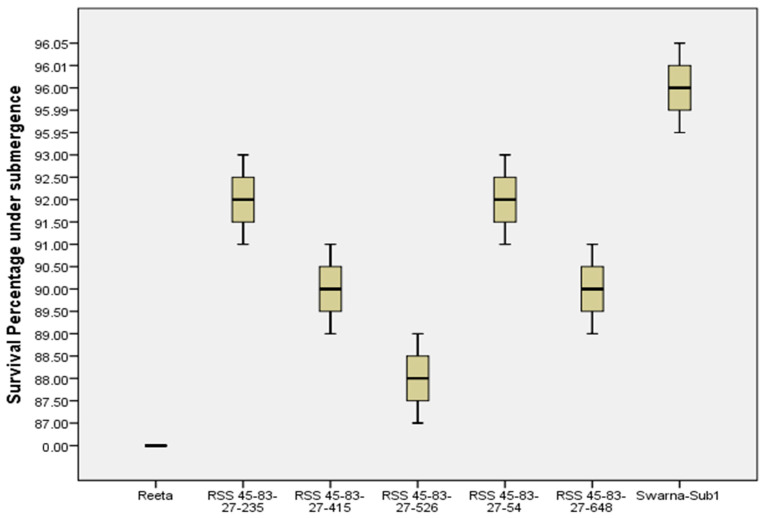
Percent of plants regenerated in the parental lines Reeta and Swarna-Sub1 and their pyramided lines carrying *Sub1* QTL under control screening facility after one week of de-submergence from 14 days of submergence stress.

**Figure 8 ijms-24-10708-f008:**
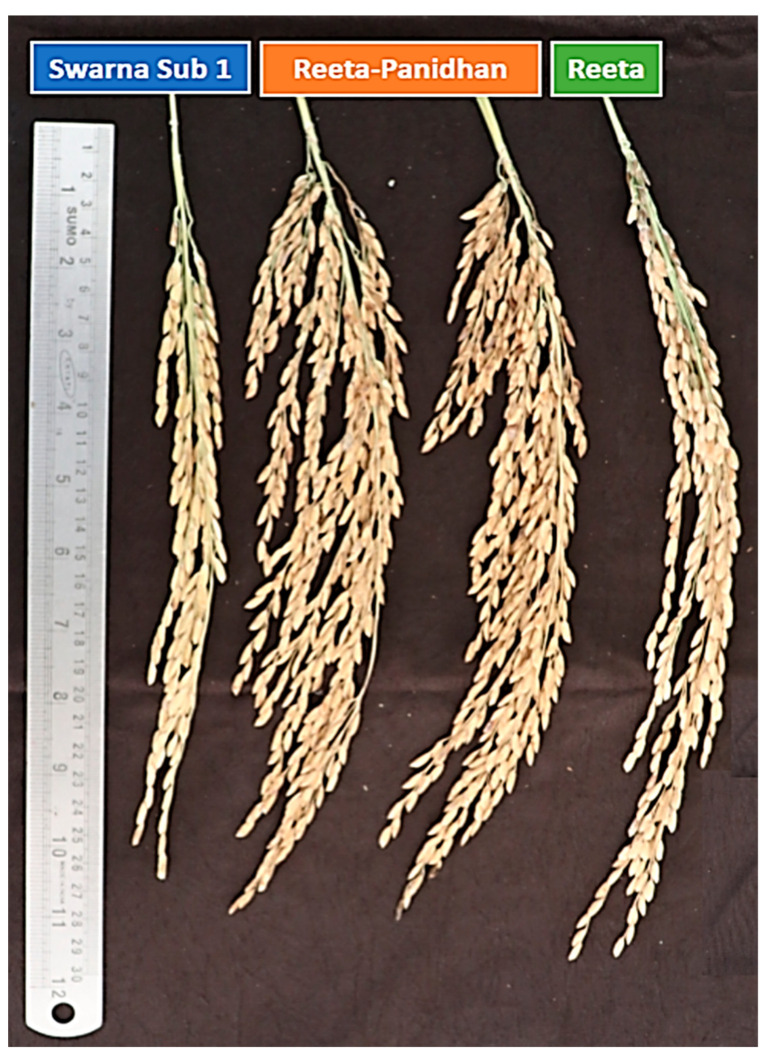
Successful introgression of *Sub1* and *Pup1* QTLs for stress resiliency and increased panicle yield in the flood-prone late-maturing rice using marker-assisted breeding. Photograph shows the panicles of parents, Reeta (right) and Swarna (left) along with the best pyramided lines (middle two) observed during the wet season, 2021.

**Figure 9 ijms-24-10708-f009:**
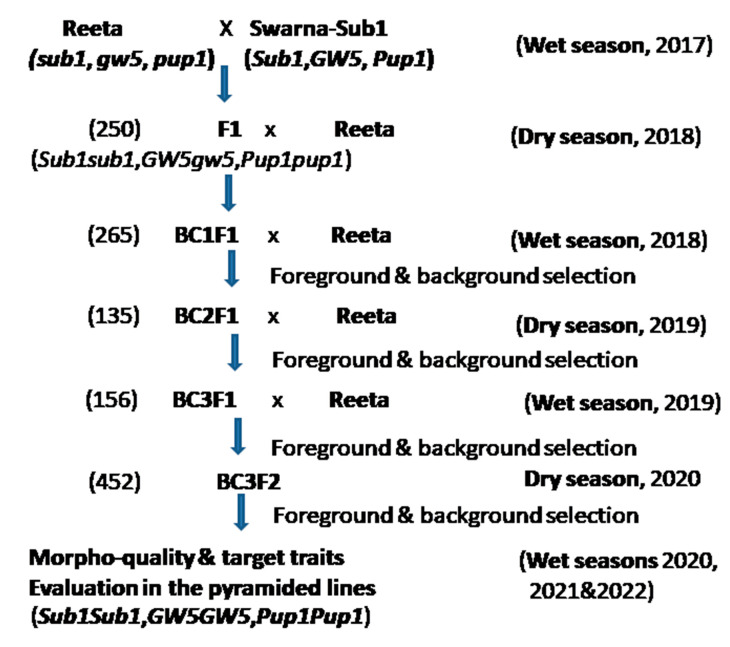
The marker-assisted breeding scheme used for the development of pyramided lines for screening of submergence, low phosphorus stress tolerance, and panicle yield component QTLs. The numerical figures within parentheses indicate the number of hybrids/lines raised in the generation.

**Table 1 ijms-24-10708-t001:** Genotyping of backcross progenies for three QTLs and recovery of recipient parent’s genome in the foreground positive backcross progenies.

Generation	No. of Plants Scored	No. of Progeny Heterozygotes for the 3 Target QTL	Expected % of Recurrent Parent Genome to Selected Backcross Plants	Average Recipient Parent Genome Content (%) in the Backcross Progenies	Maximum Genome Recovery (%) of Recipient Parent in the Selected Progenies	Chi-Square Value (χ^2^)
BC_1_F_1_	138	9	75.0	76.91	80.95	0.049
BC_2_F_1_	135	8	87.5	89.06	92.26	0.028
BC_3_F_1_	156	7	93.25	94.30	95.24	0.012

**Table 2 ijms-24-10708-t002:** Phosphorus uptake (mg/g tissue dry weight) and grain yield of the parental lines Reeta and Swarna-Sub1 and their pyramided lines grown under normal and P-deficient soils.

Scheme	Pyramided and Parental Lines	PUptake (mg/g Tissue Dry Weight)		Grain Yield (Average Single Plant in Gram)	
		Normal	Deficient	Normal	Deficient
1	RSS 45-83-27-54	0.224	0.205	25.67	22.54
2	RSS 45-83-27-235	0.218	0.212	24.25	21.55
3	RSS 45-83-27-415	0.208	0.194	24.28	22.16
4	RSS 45-83-27-526	0.232	0.216	22.56	20.68
5	RSS 45-83-27-648	0.228	0.202	22.32	20.16
6	Reeta (–ve check)	0.156	0.138	21.56	15.24
7	Swarna-Sub1 (+ve check)	0.238	0.216	22.21	20.14
CV%		9.43	10.25	8.26	9.14
CD_5%_		0.042	0.048	2.182	2.534

**Table 3 ijms-24-10708-t003:** Agro-morphologic and grain quality traits of the pyramided lines along with parents pooled over 3 seasons.

Sl. No.	Pyramided and Parental Lines	PH	DFF	NP	GN	PL	SW	SF	NPB	PW	GL	GB	H	M	HRR	AC	ASV	GC	PY
1	RSS 45-83-27-54	117	120	12	218	28.3	21.7	83.5	14.3	5.64	5.12	2.28	77	68.5	64.3	20.7	5.0	48	6.75
2	RSS 45-83-27-235	115	119	11	212	28.1	21.4	83.2	14.1	5.21	5.18	2.30	76	67.8	65.1	21.2	5.0	47	6.65
3	RSS 45-83-27-415	116	120	10	215	27.8	21.8	82.6	13.4	5.25	5.03	2.27	75	69.4	64.5	21.4	4.5	48	6.48
4	RSS 45-83-27-526	115	118	10	208	27.5	21.5	82.3	13.6	5.56	5.04	2.28	76	66.5	63.6	21.3	4.0	49	6.40
5	RSS 45-83-27-648	117	120	9	210	27.4	21.6	82.7	13.7	5.08	5.08	2.28	78	67.2	63.8	21.5	5.0	51	6.25
6	Swarna-Sub1	110	118	13.25	164	26.2	20.32	83.8	14.2	5.16	5.175	2.22	77	68.45	61.3	24.25	4.0	55.5	6.15
7	Reeta (Recipient)	116	120	9.25	191	28.4	24.58	77.6	12.4	4.28	4.97	2.32	79	70.9	65.3	20.5	5.0	47	4.38
LSD_5%_		8.74	4.35		2.94	18.3		1.72	1.07	0.515	0.71	1.86	0.13	6.58	6.85	7.64	1.883	2.45	0.275
CV%		3.08	0.85		10.24	9.82		5.85	7.82	10.13	7.24	3.26	7.68	5.24	6.76	8.21	6.731	4.35	10.72

PH: Plant height (cm); DFF: Days to 50% flowering; NP: Panicles/plant; GN: Grains/panicle; PL: Panicle length (cm); SW: 1000-seed weight (g); SF: Spikelet fertility %; NPB: No. of primary branches; PW: Panicle weight (g); GL: Grain length (mm); GB: Grain breadth (mm); H: Hulling %; M: Milling (%); HRR: Head rice recovery (%); AC: Amylose content (%); ASV: Alkali Spreading Value; GC: Gel Consistency (mm); PY: Plot yield (t/ha).

## Data Availability

The original contributions presented in the study are included in the article.

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
