# Peer review of "Transfer of Stress Resilient QTLs and Panicle Traits into the Rice Variety, Reeta through Classical and Marker-Assisted Breeding Approaches"

_ijms, 2023, doi:10.3390/ijms241310708_

Round 1

Reviewer 1 Report

Comments to the Authors

The manuscript entitled “Transfer of stress resilient QTLs and panicle traits into the rice variety, Reeta through classical and marker-assisted breeding approaches” explains the current needs of developing climate resilient rice variety. However, the manuscript has to be drastically improved and reconsider after major revision. The comments to the authors are as below. 

General comments:

The precise representation of the work and grammatical errors have to be taken care.

The authors extensively quoted self-citations; required to quote the original work and remove unnecessary self-citations.

The reference numbers mentioned towards supporting work are not matching with reference list numbers. For one reference number provided two references in several cases (For example: two references are available for reference #1 etc.). All the references should be formatted as per the journal guidelines as references have not been formatted uniformly.

The SI units of all the measurements have be to be mentioned properly.

The gene name should be italicized throughout the manuscript including references in the end of the manuscript.

The high resolution images should be provided as most of the images are not clear in content and labelling of figures and their legends as well. Statistical significance should be included wherever necessary for comparisons.

Abstract

Abstract has been written precisely, however, the fold change improvement with respect to target genes (submergence, low phosphorous and grain yield) in the performance of the introgressed lines as compared to original line should be mentioned.

Use “Recurrent parent genome” instead of “Recipient parent genome” as aforesaid terms are more popular and common than the present words.

Introduction

Authors must highlight the coverage of Reeta variety and comparative advantage of Reeta with Swarna Sub-1 as donor variety Swarna Sub-1 possessed all the target genes and even the phenotypic performance of the introgressed lines of Reeta are all most similar with Swarna Sub-1.

Materials and Methods

The Reeta variety name must be uniformly mentioned throughout the manuscript.

Mention the allelic status of all the QTLs/genes in the flow diagram of MAB scheme for easy understanding of the readers.

The “statistical analysis” should be mentioned in separate sub-heading.

Table 1 regarding the marker details have to be provided in supplementary table.

Mention the crop stage of sampling for phosphorous estimation and crop stage for sub-mergence testing.

Table 2 can be shifted to supplementary table as background recovery is represented in graphically.

Results

Figure 2 can also be shifted to supplementary material as foreground selection gel picture depicts the polymorphism between the parents in each generation.

Provide the foreground selection results in the form of table considering each QTL segregation separately and analyse the segregation ratios through chi-square test.

Crosscheck the number of the positive plants mentioned in foreground selection for each QTL/gene as some mismatched can be clearly observed in point #3.4 of results section.

Figure 8 must be improved to provide the chromosome wise recurrent parent genome recovery and legend A and B must be labelled.

In point number 3.8 subheading, mention the conditions (normal or submergence/low-phosphorous) in which the pyramided lines were evaluated for agro-morphological, yield and grain quality traits?

Figure 11 should be shifted to supplementary material as figure 7 also provides the similar details.

Result section has been written very elaborately, avoid the repetition of the content.

Discussion

Precise revising and flow of the content in discussion must be improved with supporting literature.

Conclusion

The conclusion should mention the improvement in performance through introgression of target QTLs in the form fold change compared to their original performance.

Author Response

Point-by-point response to the reviewer’s queries

Comments to the Authors

The manuscript entitled “Transfer of stress resilient QTLs and panicle traits into the rice variety, Reeta through classical and marker-assisted breeding approaches” explains the current needs of developing climate resilient rice variety. However, the manuscript has to be drastically improved and reconsider after major revision. The comments to the authors are as below. 

General comments:

Query: The precise representation of the work and grammatical errors have to be taken care.

Response: We have revised the article and due care has been given for grammatical errors.

Query: The authors extensively quoted self-citations; required to quote the original work and remove unnecessary self-citations.

Response: We have removed the unnecessary self citations.

Query: The reference numbers mentioned towards supporting work are not matching with reference list numbers. For one reference number provided two references in several cases (For example: two references are available for reference #1 etc.). All the references should be formatted as per the journal guidelines as references have not been formatted uniformly.

Response: thanks. The references are formatted as per the journal guidelines and also repeated references are removed.

Query: The SI units of all the measurements have be to be mentioned properly.

Response: All the measurements are mentioned in SI units.

Query: The gene name should be italicized throughout the manuscript including references in the end of the manuscript.

Response: The genes are now italicized throughout the manuscript.

Query: The high resolution images should be provided as most of the images are not clear in content and labelling of figures and their legends as well. Statistical significance should be included wherever necessary for comparisons.

 Response: The resolutions of images have been increased.

Abstract

Query: Abstract has been written precisely, however, the fold change improvement with respect to target genes (submergence, low phosphorous and grain yield) in the performance of the introgressed lines as compared to original line should be mentioned.

Response: The improvement with respect to target genes (submergence, low phosphorous and grain yield) in the introgressed lines compared to the recipient parent is better which is included in the revised manuscript.

Query: Use “Recurrent parent genome” instead of “Recipient parent genome” as aforesaid terms are more popular and common than the present words.

Response: Thanks. Recurrent parent genome term is instead of recipient parent genome

Introduction

Authors must highlight the coverage of Reeta variety and comparative advantage of Reeta with Swarna Sub-1 as donor variety Swarna Sub-1 possessed all the target genes and even the phenotypic performance of the introgressed lines of Reeta are all most similar with Swarna Sub-1.

Response: Reeta variety is popular with the farmers under late maturity group but the variety is sensitive to submergence stress due to flash flood. The growers are incurring a heavy loss under such situation due to the crop failure. Flash flood due to heavy rain may be occasional but occurrences are unpredictable. The panicle of Reeta variety gives more grains/panicle and bear a heavy panicle than the variety, Swarna-Sub1. However, the grains of Reeta are bold type with more gain width than Swarna-Sub1.

We have incorporated the superior traits of Reeta variety under introduction section.

Materials and Methods

Query:The Reeta variety name must be uniformly mentioned throughout the manuscript.

Response: Thanks. We have used Reeta in the revised manuscript.

Query: Mention the allelic status of all the QTLs/genes in the flow diagram of MAB scheme for easy understanding of the readers.

Response: The allelic status of all the QTLs/genes are now depicted in the flow diagram of MAB scheme.

Query: The “statistical analysis” should be mentioned in separate sub-heading.

Response: We have now included all the statistical analysis under separate sub-heading.

Query: Table 1 regarding the marker details have to be provided in supplementary table.

Response: Table1 is shifted  and placed as a the supplementary table.

Query: Mention the crop stage of sampling for phosphorous estimation and crop stage for sub-mergence testing.

Response: The crop growth stage for phosphorous estimation and sub-mergence testing are included in the revised manuscript.

Query: Table 2 can be shifted to supplementary table as background recovery is represented in graphically.

Response: Table 2 is also shifted  as a the supplementary table.

Results

Query: Figure 2 can also be shifted to supplementary material as foreground selection gel picture depicts the polymorphism between the parents in each generation.

Response: Figure 2 is also moved  to the supplementary Figures.

Query: Provide the foreground selection results in the form of table considering each QTL segregation separately and analyse the segregation ratios through chi-square test.

Response: Foreground selection results are presented in a table and the segregation ratios are analyzed through chi-square test.

Query: Crosscheck the number of the positive plants mentioned in foreground selection for each QTL/gene as some mismatched can be clearly observed in point #3.4 of results section.

Response: Thanks. We have checked the mismatch.

Query: Figure 8 must be improved to provide the chromosome wise recurrent parent genome recovery and legend A and B must be labelled.

Response: This is a software generated figure. The legend A and B are labeled now in the revised manuscript.

Query: In point number 3.8 subheading, mention the conditions (normal or submergence/low-phosphorous) in which the pyramided lines were evaluated for agro-morphological, yield and grain quality traits?

Response: We have now included the conditions of testing under material methods and results section.

Query: Figure 11 should be shifted to supplementary material as figure 7 also provides the similar details.

Response: Figure 11 is also moved  to the supplementary Figures.

Query: Result section has been written very elaborately, avoid the repetition of the content.

Response: Please see the revised manuscript for the changes.

Discussion

Precise revising and flow of the content in discussion must be improved with supporting literature.

Response: We have revised the discussion section again.

Conclusion

The conclusion should mention the improvement in performance through introgression of target QTLs in the form fold change compared to their original performance.

Response: The improvement with respect to target genes (submergence, low phosphorous and grain yield) in the introgressed lines compared to the recipient parent is better which is included in the revised manuscript.

Reviewer 2 Report

The authors reported a successful backcross breeding program in rice using marker assisted selection. The major quantitative trait loci (QTLs) for submergence tolerance, low phosphorus stress tolerance and narrow grains were transferred from Swarna-sub1 (donor) to Reeta (recipient). Five lines homozygous for the three QTLs were shown to yield better than the recipient parent and remain similar in other desirable traits to the recipient parent. Overall, I think the manuscript is quite straightforward and I have only few minor comments to suggest.

Section 2.1, Line 3. Please check for spelling consistency for the variety name, Reeta vs Rita.

Section 2.2, Line 2. Please correct the spelling error in “APCR”.

Figure 1. The figure legend is incomplete as part of it bleeds into the main text. Please correct the figure legend formatting.

Table 1. What does position mean here? Does it mean the position of the markers in the genome? If yes, then bp does not seem right. Or, does it mean the amplified product length?

Figure 7. The fonts in this figure are illegible, so please enlarge the fonts and provide a better resolved figure. The subplot labels (A, B, C) are missing. The bar plot is unnecessary, and it is easier to just show the percentage in parentheses after the line names in the dendrogram.

Figure 8. The fonts in this figure are also illegible, so please enlarge the fonts and provide a better resolved figure. Please also modify the figure to scale the vertical axis according to either physical or genetic positions of the markers. It is quite misleading in its current form.

Table 5. Spikelet sterility is redundant as it is simply 100 – fertility. Please remove either column.

Figure 11. The fonts in this figure are completely illegible, so please enlarge the fonts and provide a better resolved figure.

Table 6. Many of the columns in this figure are duplicated from Table 5. Please either remove these duplicated columns, or merge the two tables together.

Author Response

Point-by-point response to the reviewer’s queries

The authors reported a successful backcross breeding program in rice using marker assisted selection. The major quantitative trait loci (QTLs) for submergence tolerance, low phosphorus stress tolerance and narrow grains were transferred from Swarna-sub1 (donor) to Reeta (recipient). Five lines homozygous for the three QTLs were shown to yield better than the recipient parent and remain similar in other desirable traits to the recipient parent. Overall, I think the manuscript is quite straightforward and I have only few minor comments to suggest.

Query: Section 2.1, Line 3. Please check for spelling consistency for the variety name, Reeta vs Rita.

Reponse: Thanks. We have corrected the mistake.

Query: Section 2.2, Line 2. Please correct the spelling error in “APCR”.

Reponse: Thanks. We have corrected the mistake.

Query: Figure 1. The figure legend is incomplete as part of it bleeds into the main text. Please correct the figure legend formatting.

Response: Now gap is provided between figure legend and the main text.

Query: Table 1. What does position mean here? Does it mean the position of the markers in the genome? If yes, then bp does not seem right. Or, does it mean the amplified product length?

Response: Position in the table 1 is for the markers position on the chromosome.

Query: Figure 7. The fonts in this figure are illegible, so please enlarge the fonts and provide a better resolved figure. The subplot labels (A, B, C) are missing. The bar plot is unnecessary, and it is easier to just show the percentage in parentheses after the line names in the dendrogram.

Response: we have increased the resolution of the figure. We have now increased th resolution of the the figures.

Query: Figure 8. The fonts in this figure are also illegible, so please enlarge the fonts and provide a better resolved figure. Please also modify the figure to scale the vertical axis according to either physical or genetic positions of the markers. It is quite misleading in its current form.

Response: This is a software generated figure and possibility of improvement is less.

Query: Table 5. Spikelet sterility is redundant as it is simply 100 – fertility. Please remove either column.

Response. Thanks. We are removing the column for spikelet sterility.

Query:  Figure 11. The fonts in this figure are completely illegible, so please enlarge the fonts and provide a better resolved figure.

Response: We have increased the figure resolution.

Table 6. Many of the columns in this figure are duplicated from Table 5. Please either remove these duplicated columns, or merge the two tables together.

Response: As the number of traits are more, so we splitted the table into 2 tables. Now, we are merging both the Tables.

Round 2

Reviewer 1 Report

The authors have provided the point-by-point response to all the queries raised during last review process. Now, the manuscript has been improved drastically and can be accepted for publication in the current stage of peer-review process.